# The Anti-Rotaviral Activity of Low Molecular Weight and Non-Proteinaceous Substance from *Bifidobacterium longum* BORI Cell Extract

**DOI:** 10.3390/microorganisms7040108

**Published:** 2019-04-23

**Authors:** Yeo Ok Han, Yunju Jeong, Hyun Ju You, Seockmo Ku, Geun Eog Ji, Myeong Soo Park

**Affiliations:** 1Department of Food and Nutrition, College of Human Ecology, Seoul National University, Seoul 08826, Korea; skypalice@naver.com (Y.O.H.); tanklov0@snu.ac.kr (Y.J.); dhlover1@snu.ac.kr (H.J.Y.); geji@snu.ac.kr (G.E.J.); 2Research Center, BIFIDO Co., Ltd., Hongcheon 25117, Korea; 3Institute of Health and Environment, Graduate School of Public Health, Seoul National University, Seoul 08826, Korea; 4Fermentation Science Program, School of Agriculture, College of Basic and Applied Sciences, Middle Tennessee State University, Murfreesboro, TN 37132, USA; seockmo.ku@mtsu.edu

**Keywords:** rotavirus infection, *Bifidobacterium longum* BORI, probiotics, cytopathic effect, real time cell analysis

## Abstract

Rotavirus infection is the most common diarrheal disease worldwide in children under five years of age, and it often results in death. In recent years, research on the relationship between rotavirus and probiotics has shown that probiotics are effective against diarrhea. A clinical trial has reported that *Bifidobacterium longum* BORI reduced diarrhea induced by rotavirus. The present work investigated the anti-rotaviral effect of *B. longum* BORI by cytopathic effect observation and real time cell analyses. Our study found that *B. longum* BORI showed strong anti-rotaviral effect when incubated with MA104 cells prior to viral infection, suggesting that the probiotic does in fact interfere with the interaction of viruses and host cells. It is believed that the efficacy is due to low-molecular weight and non-protein components derived from *B. longum* BORI. This discovery can help broaden the industrial application of *B. longum* BORI, which has been proven to be a safe and effective probiotic.

## 1. Introduction

Rotavirus infection, the primary cause of death among children under five years of age, is caused by rotaviruses belonging to double-stranded RNA viruses, and presents symptoms including vomiting, fever, diarrhea, and dehydration [1]. It is estimated that one-third of children under five years of age who are hospitalized for diarrhea have rotavirus infections, regardless of whether they live in advanced or developing countries [2,3]. This condition is serious, as infants with severe dehydration due to diarrhea can die.

Because there was no proper treatment in the 1990’s, RotaShield, an oral live vaccine derived from monkey-type rotavirus, was developed in order to prevent the infection in advance; it was approved by the US FDA in 1998 [4]. However, risks of complication with RotaShield such as intussusception and bowel obstruction were reported, and in 1999 the vaccine manufacturer voluntarily withdrew the license from the market [4]. Then, in 2006 and 2008, two live oral rotavirus vaccines called RotaTeq and Rotarix, were approved by the US FDA for the prevention of rotavirus gastroenteritis in infants [5]. These two vaccines are widely used globally as there is less risk of intussusception with the vaccines than with RotaShield and even WHO has recommended including these vaccines in national immunization schedules [5]. Nevertheless, the high price of the two vaccines makes it difficult for developing countries and regions such as West Africa and Asia to acquire them. In 2013, deaths from rotavirus were 215,000 globally and 41% of them occurred in Asian countries. Since the vaccine was introduced, only in eight countries, the morbidity and mortality due to rotavirus infection is still high in Asia [6]. Therefore, there is a need for alternative preventive measures that are economical and easy to use.

Probiotics have been found to be effective against diarrhea, and researchers are beginning to study their effects on rotavirus. *Bifidobacterium adolescentis* and *Lactobacillus casei* have been shown to block the adherence of rotavirus to the MA104 cells [7]. Another study revealed that milk fermented with *B. breve* C50, *Streptococcus thermophilus* 065, and combined with prebiotics prevents rotavirus-induced diarrhea when fed to suckling rats [8]. In another study, *B. breve* M016V was shown to have a protective effect on the rotavirus infection model [9]. In addition, we observed that the duration of diarrhea was significantly reduced by feeding *B. longum* BORI and *Lactobacillus acidophilus* AD031 to rotavirus-infected infants [10].

Therefore, we conducted the present study to reveal how the probiotic bacteria contributes to anti-rotaviral activity. We focused on *B. longum* BORI, which showed the reduction of diarrhea in the preceding study [10].

## 2. Materials and Methods

### 2.1. Cells, Viruses, and Bacteria

In this study, MA104 (ATCC, Manassas, VA, USA) from the African green monkey’s kidney epithelial cell was used to propagate rotavirus. The Wa strain (ATCC, Manassas, VA, USA), human rotavirus A, was used to infect the MA104 cells. Then, the MA104 cells were cultured in Dulbecco’s modified eagle’s medium (DMEM, Thermo Fisher Scientific, Waltham, MA, USA), supplemented with 10% (v/v) fetal bovine serum (Thermo Fisher Scientific, Waltham, MA, USA), and 1% (v/v) antibiotics (Thermo Fisher Scientific, Waltham, MA, USA) and sub-cultured by detaching with 0.25% (v/v) of trypsin-EDTA (Thermo Fisher Scientific, Waltham, MA, USA) when 80% was confluent with the flask. Cells were maintained in an incubator saturated with 5% CO_2_. The tested bacteria were *B. longum* RD43, RD61, RD65, RD69, RD118, RD138, and BORI. All bacteria were isolated from human feces of healthy adults and infants who lived in Chuncheon, South Korea between 1995 and 1998, identified with 16S rRNA sequencing and cultured with MRS (De Man, Rogosa and Sharpe, Becton Dickson, Franklin Lakes, NJ, USA) broth medium at 37 °C. For the initial screening, lyophilized powder of each strain was used. The concentration of *B. longum* strains for screening was 3 µg/mL.

### 2.2. Preparation of Cell Extract of Tested Bacteria

*B. longum* BORI was cultured in MRS broth medium at 37 °C for 18 h and collected by centrifugation at 7000 rpm for 1h. To acquire the cell extract of bacteria, the collated pellet was washed with phosphate buffered saline (PBS) to remove the MRS broth medium, sonicated twice for 15 min on ice, and centrifugated at 15,000 rpm for 30 min. The supernatant of sonicated and centrifugated cells were filtered with a 0.45 µm pore size syringe filter (Pall Corporation, Port Washington, NY, USA) and used as the bacterial whole cell extract (BCE). To prepare the enzyme and heat-treated BCE, the mixture was heated at 100 °C for 20 min or autoclaved at 121 °C for 15 min and proteolytic enzymes such as proteinase K (Sigma-Aldrich, St. Louis, MO, USA) and trypsin (Sigma-Aldrich, St. Louis, MO, USA) was treated to BCE according to the manufacturer’s enzymatic assay. Finally, the fractions of cell extract divided by the molecular weight were prepared by membrane filtration as appropriate (Sigma-Aldrich, St. Louis, MO, USA).

### 2.3. Antiviral Assay

MA104 cells were detached from the flask by treating with 0.25% of trypsin-EDTA and washing with DMEM medium including 10% FBS. Wa virus stock solution (10^6.41^TCID50/mL) was diluted to one-hundredth to the DMEM medium, including 5 µg/mL trypsin (Thermo Fisher Scientific, Waltham, MA, USA). With 50 μL of the diluted virus solution, 50 μL of 15,000 MA104 cells, and 100 μL of BCE was mixed and incubated in 96 well plates for 5 days at 37 °C and 5% CO_2_ condition dependent on the experimental condition. The final Wa virus concentration was 6425.99TCID50/mL. To test whether the BCE directly affected the viruses, MA104 cells were added to the samples after the diluted virus solution and BCE mixture were incubated for 30, 60, and 90 min at 37 °C. To test pre-treatment efficacy, the MA104 cells were treated with BCE for 30, 60, and 90 min before the viral infection emerged. To measure the activity against rotavirus infection, cytopathic effect (CPE), rounding and detachment of cells, and plaque formation were all observed through the microscope. In addition to CPE, a colorimetric assay, MTT assay was performed to quantify the antiviral activity of BCE. The inhibition of infection was expressed as the percent (%) of wells with uninfected cells among the total experimented wells. The real time antiviral effect was observed using the xCelligence RTCA SP system (ACEA Bioscience, San Diego, CA, USA), a real-time cell analyzer. The inhibition of infection was then determined by measuring the cell index (CI), which represents the cell status as relative change in measured electrical impedance to represent cell status. When cells are detached from the well due to lysis, the CI is zero.

### 2.4. Colorimetric Trypsin Inhibitor Assay

The colorimetric trypsin inhibitor assay was performed to evaluate trypsin inhibitory effect of BCE following the methods of Kerry et al [11]. Specifically, the following equation was used for assessing the inhibited trypsin unit;
Inhibited trypsin unit (U) = 100 × (A410c − A410s)
where A410c and A410s are the absorbance of control and absorbance of the sample at 410 nm, respectively. One trypsin inhibitor unit is defined as a decrease of 0.01 absorbance units at 410 nm per 10 mL reaction mixture.

### 2.5. Statistical Analysis

Data are expressed as the mean ± standard deviation (SD). The one-way analysis of variance (ANOVA) with Tukey’s post hoc test was implemented for multiple comparisons. All analyses were done with Prism 7 software (GraphPad Software, San Diego, CA, USA).

## 3. Results

### 3.1. Anti-Rotaviral Effect of BCE on Rotavirus

To select anti-rotaviral bacteria, 7 *Bifidobacterium longum* strains, that had been isolated from the feces of infants were screened on MA104 cells. Among the strains, *B. longum* BORI showed the most potent efficacy for inhibiting rotavirus infection (Table 1).

To investigate whether BCE has more of an effect on rotavirus itself, BCE was treated with the Wa virus before the MA104 cells were infected. Results showed that under 3 mg/mL of BCE the inhibition effect was dependent on dose and incubation time (Figure 1A). The rotavirus infection of the host cells is generally known to be enhanced by the action of host trypsin on rotavirus [12]; thus, in this study, colorimetric trypsin inhibitor assay was performed and BCE showed dose dependent trypsin inhibitory efficacy (Figure 1B).

In order to determine whether BCE is associated with the interaction between virus and host cells as well as influencing the virus itself, BCE was added to MA104 cells before virus infection. As a result, the inhibition effect was more potent at a relatively lower concentration compared to the co-treatment of BCE and the virus (Figure 2).

### 3.2. Anti-Rotaviral Effect of Heat and Proteolytic Enzyme Treated BCE

Heat and proteolytic enzymes were applied to the BCE prior to antiviral assay for investigating whether the protein portion of the BCE is involved in anti-rotaviral efficacy. As a result, it was confirmed that the antiviral ability of BCE was maintained regardless of the heat treatment (Figure 3). In addition, we observed that BCE activity was maintained without being influenced by enzymes of Proteinase K or trypsin (Table 2).

### 3.3. The Antiviral Effect of BCE Fractions Divided by the Molecular Weight

To determine which part of the bacteria exerted the anti-rotaviral effect, we separated BCE into the low molecular weight fraction (<10 kDa) and high molecular fraction (≥ 10 kDa) by using a 10 kDa cutoff filter. When the tested samples were pre-treated to MA104 cells for 90 min before infection, the real-time cell profile showed that there was an anti-rotaviral effect in only the low molecular weight fraction (Figure 4). There was no effect in the high molecular weight fraction, and the whole cell extract showed an intermediate effect between low and high molecular weight fractions.

## 4. Discussion

*B. longum* BORI, isolated from healthy infants, has been proven to be safe and has been used for a decade as a commercial probiotic product world-wide [13]. Recently, it was discovered that oral administration of *B. longum* BORI could shorten the duration of diarrhea induced by rotavirus infection [10].

In order to find a component that is involved in antiviral activity of *B. longum* BORI, further studies were conducted with cell extract of the bacterium (BCE). First, we confirmed that the anti-rotaviral effect of BCE through CPE observation was dependent on time and concentration. Interestingly, BCE could reduce the trypsin activity, which is known for inducing rotaviral infection through cleavage of VP4 capsid protein to VP5* and VP8* for adhesion to host cells [14]. Hence, BCE seems to contribute to the inhibitory efficacy of rotaviral infection indirectly by inhibiting trypsin. When MA104 cells were treated with BCE before rotavirus infection, the inhibition effect of viral infection was stronger than when rotavirus was treated with BCE before infection of MA104 cells and after incubation for a period of time. These results suggest that BCE is more effective in inhibiting viral infection by acting on the host cell than affecting the virus itself.

We performed real time cell analysis with the Xcelligence instrument in addition to microscopy observation to verify the CPE. Real time cell analysis is a method of monitoring and recording cells in real time, in contrast to the end-point assay, which is measured at a specific time [15]. When cells are detached from the well, showing a deformed morphology, the cell index (CI) decreases [15]. In general, the end-point assay of the MA104 cell line was performed with a 5-day incubation, but we were able to confirm the virus-mediated cytopathogenecity in a shorter time with the real time cell analysis method. Similar to microscopic observation, pre-treating MA104 cells with BCE prior to infection inhibited viral infection and allowed it to maintain a higher CI value than the negative control cells that had not been infected. Intriguingly, this effect was maintained even in the heat-treated BCE. Antiviral efficacy was also maintained when proteases such as trypsin and proteinase K were applied to BCE. These results suggest that the anti-rotaviral efficacy of BCE is due to non-protein substances. In order to determine which part of the bacteria is involved in the anti-rotaviral effect, we implemented real time cell analysis with two fractions of cell extract based on the molecular weight. The low molecular weight fraction of under 10 kDa seemed to be more effective than whole cell extract based on the real time cellular profile. The results suggest that low-molecular weight and non-protein components of *B. longum* BORI play a major role in antiviral activity.

Rotavirus infection is reported to consist of a multistep process involving VP4 and VP7 capsid proteins of rotavirus, along with host cellular sialic acid, heat shock cognate 71 kDa (Hsc70), and integrin [16]. Several studies on the anti-rotaviral efficacy of probiotics have assumed that proteins derived from probiotic bacteria play an important role in the interaction between viruses and host cells. For example, Salas-Cárdenas et. al. reported that protein rich extracts of *Bifidobacterium spp.* interfered with rotavirus adhesion on MA104 cells by binding b3-integrins and Hsc70 [17]. In addition, Chenoll et al. found that proteinaceous substance, an 11-amino acid peptide, from supernatant of *B. longum* CECT 7210 inhibit the replication of rotavirus [18]. In addition, protein-based metabolite of *Bifidobacterium adolescentis* showed antiviral activity against rotavirus infection in the research of Olaya et al. [19]. A high molecular weight (76 kDa) protein, purified from *Bifidobacterium breve* K-110, showed inhibitory activity toward the infection of rotavirus [20], however, the genetic information of this protein is not available. A number of studies have reported that protein-based compounds exhibit anti-rotaviral activity but it was difficult to find a study about non-protein components from probiotic strain showing an anti-rotaviral effect. As is well known, proteins are fragile substances that are denatured by heat and degraded by gastric proteolytic enzymes, thus there are restrictions on industrial applications such as transport and storage that ensure specified conditions. However, our results could show that non-protein substances interfere with the interactions between viruses and host cells. In addition, *B. longum* BORI was recently approved as a new dietary ingredient by the US FDA based on the robust safety data such as antibiotic resistance and hemolytic effect tests (NDI 1082). Therefore, when developed commercially, the substance found in this study can be used more easily than other proteinaceous materials derived from probiotic bacteria.

Taken together, we discovered that low-molecular weight (<10 kDa) and non-protein components derived from *B. longum* BORI play important roles in anti-rotaviral effects. The industrial application of *B. longum* BORI, which has been proven a safe and probiotic bacterial strain, could be expanded through the continuation of the present work. Further studies are needed to characterize and identify this active compound in order to reveal its action mechanism.

## Figures and Tables

**Figure 1 microorganisms-07-00108-f001:**
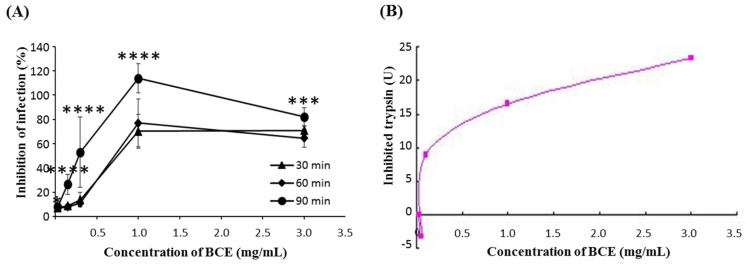
The effects of bacterial whole cell extract (BCE) on rotavirus infection. (**A**) BCE and rotavirus were incubated for 30, 60, 90 min together prior to treatment. The single, triple asterisk and quadruple asterisk indicate significant difference of the inhibitory effect of infection of BCE exposure to the cells among different time points (* *p* < 0.05, *** *p* < 0.001, **** *p* < 0.0001). Data are expressed as the mean ± SD (*n* = 8). (**B**) Calorimetric trypsin inhibitor assay was performed with various concentrations of BCE.

**Figure 2 microorganisms-07-00108-f002:**
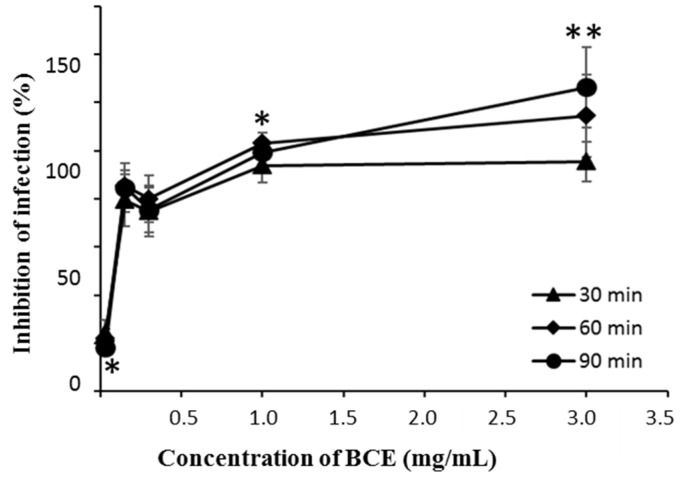
The effects of BCE on the rotavirus infection in MA104 cells according to time and dose. The BCE was pretreated to MA104 cells for 30, 60, and 90 min before rotavirus infection. The infected MA104 cells were incubated at 37 °C for 5 days. The single and double asterisk indicate significant difference of the inhibitory effect of infection of BCE among different time points (* *p* < 0.05, ** *p* < 0.01). Data are expressed as the mean ± SD (*n* = 8).

**Figure 3 microorganisms-07-00108-f003:**
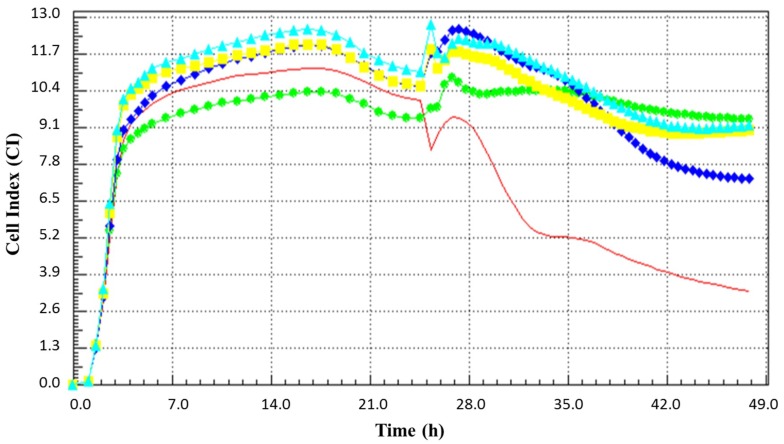
The real time cell monitoring profile of MA104 cells treated with 0.25 mg/mL of BCE prior to the rotavirus infection. The blue diamond line is the BCE, the light blue triangle line is the autoclaved BCE, the yellow square line is the heat treated BCE, the red line represents infected cells, and green circle line represents non-infected cells.

**Figure 4 microorganisms-07-00108-f004:**
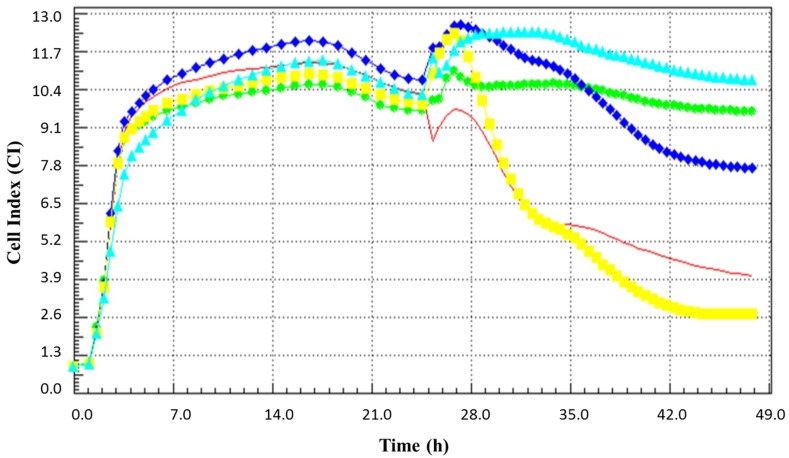
The real time cell monitoring profile of MA104 cells treated with the fraction of BCE based on the molecular weight prior to the rotavirus infection. The blue diamond line is the cell extract of B. longum BORI, the light blue triangle line is the low molecular weight (<10 kDa) fraction of cell extract, the yellow square line is the high molecular weight fraction (≥10 kDa) of cell extract, the red line represents infected cells, and green circle line represents non-infected cells.

**Table 1 microorganisms-07-00108-t001:** The antiviral effect of lyophilized *Bifidobacterium* 3 µg/mL on rotavirus infected MA104 cells.

Strain	The Number of Infected Wells (per 8 Wells)	The Inhibition of Infection (%)
*B. longum* RD43	2/8	75
*B. longum* RD61	3/8	62.5
*B. longum* RD65	6/8	25
*B. longum* RD69	2/8	75
*B. longum* RD118	7/8	12.5
*B. longum* RD138	4/8	50
*B. longum* BORI	1/8	87.5

**Table 2 microorganisms-07-00108-t002:** The antiviral effects of enzyme treatment of BCE prior to infection of MA104 cells.

Enzyme Treatment	The Number of Infected Wells (per 8 Wells)	Inhibition Rate (%)
Trypsin + BCE	2/8	75
Proteinase K + BCE	0/8	100
BCE only (0.25 mg/mL)	0/8	100
Heat inactivated Trypsin and Proteinase K	8/8	0
Negative control	8/8	0

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
