# Peer review of "The Anti-Rotaviral Activity of Low Molecular Weight and Non-Proteinaceous Substance from Bifidobacterium longum BORI Cell Extract"

_microorganisms, 2019, doi:10.3390/microorganisms7040108_

Round 1

Reviewer 1 Report

The authors present new findings that have the potential to advance interventions against rotavirus infection. The manuscript requires moderate English editing and more detailed descriptions of prior research and methodology.

Abstract:

Lines 25, 39, 42: specify 'years' when describing age.

26: A clinical trials

32: Plural: 'low-molecular weight, non-protein components'

Introduction:

My cursory PubMed search shows multiple more recent articles that can be cited regarding rotavirus and diarrhea prevalence and treatment in children.

Line 46: Revise sentence since current treatments can be considered proper.

Line 67: Cite the specific study.

Materials and Methods:

Line 78: Explain the use of infants for fecal samples (how were they chosen; breastfed or formula-fed; age; health status; informed consent from parents, etc.

Line 100: correct the spaces and add : as appropriate.

Results:

Line 158: Change to: "In addition, we observed..."

Figures 3 and 4: The darker blue line... the light blue line...yellow line...

Line 171: correct 'Figure' 4 and "there was no effect..."

Discussion:

Line 182: world-wide

Line 189: add reference

Line 194: remove "of"

Line 209: The results suggest...components...play (remove will)

Line 211: a multistep process

Line 223: exhibit

Line 224, 233: components

Line 227: remove 'obviously'

Lne 234: effects

Line 236: remove 'discovery' and use 'continuation'

Line 236-7: incomplete sentence

Author Response

Reviewer 1:

Comments to the Author

The authors present new findings that have the potential to advance interventions against rotavirus infection. The manuscript requires moderate English editing and more detailed descriptions of prior research and methodology

Letter to reviewer

We would like to express our appreciation for the reviewer’s thoughtful suggestions. We were able to revised and improve the paper as a result of your valuable feedback.

Sincerely,

Comments in Abstract:

Lines 25, 39, 42: specify 'years' when describing age.

We have corrected the expression. Please see line #25, #39, and #41-2.

26: A clinical trials

We have corrected the expression. Please see line #26.

32: Plural: 'low-molecular weight, non-protein components'

We have corrected the expression. Please see line #31-2.

Comments in Introduction:

My cursory PubMed search shows multiple more recent articles that can be cited regarding rotavirus and diarrhea prevalence and treatment in children.

We have added contents regarding rotavirus and diarrhea prevalence and treatment in children. Please see #54-7.

(Added contents)

In 2013, deaths from rotavirus were 215,000 globally and 41% of them occur in Asian countries. Since the vaccine was introduced only in 8 countries, the morbidity and mortality due to rotavirus infection is still high in Asia.

Line 46: Revise sentence since current treatments can be considered proper.

We tried to explain why the RotaShield, an oral vaccine, was developed but the sentence did not seem to be appropriate as you commented, so we modified the following sentence. Please see line #45-7.

Line 67: Cite the specific study.

We have added the citation. Please see line #68-9.

Comments in Materials and Methods:

Line 78: Explain the use of infants for fecal samples (how were they chosen; breastfed or formula-fed; age; health status; informed consent from parents, etc.

The tested bacteria were B. longum RD43, RD61, RD65, RD69, RD118, RD138, and BORI. All bacteria were isolated from human feces of healthy adults and infants who lived in Chuncheon, South Korea between 1995 and 1998, identified with 16S rRNA sequencing and cultured with MRS broth medium at 37. We have corrected the manuscript with this information in materials and methods section. Please see line #79-82.

Line 100: correct the spaces and add: as appropriate.

We have corrected as you commented. Please see line #95-6.

Comments in Results:

Line 158: Change to: "In addition, we observed..."

We have corrected as you commented. Please see line #161.

Figures 3 and 4: The darker blue line... the light blue line...yellow line...

We have corrected as you commented. Please see line #163-6 and #177-81.

Line 171: correct 'Figure' 4 and "there was no effect..."

We have corrected as you commented. Please see line #173.

Comments in Discussion:

Line 182: world-wide

We have corrected as you commented. Please see line #184.

Line 189: add reference

We have added the citation. Please see line #190-1.

Line 194: remove "of"

We have corrected as you commented. Please see line #195-6.

Line 209: The results suggest...components...play (remove will)

We have corrected as you commented. Please see line #211-2.

Line 211: a multistep process

We have corrected as you commented. Please see line #213.

Line 223: exhibit

We have corrected as you commented. Please see line #225.

Line 224, 233: components

We have corrected as you commented. Please see line #226 and #236.

Line 227: remove 'obviously'

We have corrected as you commented. Please see line #229-30.

Line 234: effects

We have corrected as you commented. Please see line #236.

Line 236: remove 'discovery' and use 'continuation'

We have corrected as you commented. Please see line #238.

Line 236-7: incomplete sentence

We have corrected and completed the sentence. Please see line #238-9.

Reviewer 2 Report

In the manuscript entitled: “ The anti-rotaviral activity of low molecular weight and non-proteinaceous substance from Bifidobacterium longumBORI cell extract” by Han and colleagues, the authors investigated the mechanisms underlying the anti-rotaviral activity of Bifidobacterium longumBORI. In a previous publication was illustrated that from Bifidobacterium longumBORI reduce diarrhea in rotavirus-infected infants (Park et al. Nutrients 2017). Then, the final aim would be used Bifidobacterium longumBORI as pre-treatment in areas or by populations with high risk of rotavirus infections. 

While the topic of the manuscript is scientifically interesting, the analyses and results are not clearly presented and the conclusions drawn by the authors are not entirely supported by the data that are presented. For instance, the anti-viral effect of the non-proteic fraction of the BCE is not supported by the figure 3 and the table 2, where the anti-viral effect of the BCE is lost in the heat- treated BCE and by the trypsin and Proteinase K treatment.

Globally, the experimental design is weak and lack important controls. Moreover, the authors did not describe or comment the reproducibility of the results. 

Several critical concerns are listed as follows.

-       In the sentence line 157 authors said “antiviral ability of BCE was maintained regardless of the heat treatment”. However, in the figure 3, the line in yellow correspond to the heat-treated BCE and reach levels equivalents to the infected cells-not treated with BCE. Could the authors comments on it?

-      In the table 2 is not clear to which experimental set up correspond the “heat inactivated” treatment of the BCE. In any case, the combine treatment of the BCE, the trypsin and Proteinase K treatment also reduce the antiviral effect of the BCE. Could the authors comments on it?

-       In the discussion the authors stated “found that a low molecular weight fraction of under 10 kDa was more effective than whole cell extract” nevertheless in the figure 4 there is not any significant analysis done.  A commentary of the authors would be appreciated.

-      The study missed the reproducibility of their experiments. 

o  Table 1: 8 wells were used as duplicated. It is missing what it is the difference in the wells. If they are only technical duplicates, why some wells get infected and some not?. Equivalent comments for Figure 1a and figure 2: “n=3”. 

o  Figure 1B, Figure 3 and figure 4 lack of any ecartype, is it meaning that they did only 1 well or test? How many times they repeated the experiment? 

-      Regarding the percentage of inhibition of infections, represented with a Y-axis higher of 100% is not accurate. 

-      Figure 1: the authors do not comment the results regarding the high concentrations of BCE. 

Specific comments: 

-      The colours of the graphics are not compatible with colour blind people. The authors should change the colours. 

-      Moreover, the authors should to include explanations about the sign used in the graphs for the different groups in the figure or in the legend figure, to avoid the colour-print of the paper. In the actual version, they only referred to the colours in the figure legend. 

-      In the lines 99 and 100 the units looks wrong: “50 L of virus solution”, is it correct?

-      In the “antiviral assay” concentrations should be clarified for the virus solution, and for the components of the final mix, rather than dilutions or volumes. 

-      Figure 1 legend: “treatment” should be change by “exposure to the cells” or something equivalent. 

-      Globally, the strategy to illustrate significant comparisons in figure 1A, figure 2 is not clear. Authors could improve it. 

-      Line 146: “treated” should be change by “add” or something similar.

-      Line 171: type for “Figure”

-       Line 203: “and allowed it to maintain a higher CI value than the negative control cells that had notbeen infected”

Author Response

Reviewer #2:

In the manuscript entitled: “The anti-rotaviral activity of low molecular weight and non-proteinaceous substance from Bifidobacterium longumBORI cell extract” by Han and colleagues, the authors investigated the mechanisms underlying the anti-rotaviral activity of Bifidobacterium longumBORI. In a previous publication was illustrated that from Bifidobacterium longumBORI reduce diarrhea in rotavirus-infected infants (Park et al. Nutrients 2017). Then, the final aim would be used Bifidobacterium longumBORI as pre-treatment in areas or by populations with high risk of rotavirus infections.

While the topic of the manuscript is scientifically interesting, the analyses and results are not clearly presented and the conclusions drawn by the authors are not entirely supported by the data that are presented. For instance, the anti-viral effect of the non-proteic fraction of the BCE is not supported by the figure 3 and the table 2, where the anti-viral effect of the BCE is lost in the heat- treated BCE and by the trypsin and Proteinase K treatment.

Globally, the experimental design is weak and lack important controls. Moreover, the authors did not describe or comment the reproducibility of the results.

Letter to reviewer

We have addressed the specific comments on a point-by-point basis and modified the manuscript appropriately, as referred below. We thank the reviewer for the helpful feedback and assure the reviewer that we take all manuscript revision, particularly for Microorganisms, quite seriously. We found that the figure 3 was wrong figure, so we corrected the error in the revised manuscript. We are sorry for the confusion caused by the figure error. The original submission has been revised so that in concisely presents our significant research of The anti-rotaviral activity of low molecular weight and non-proteinaceous substance from Bifidobacterium longumBORI cell extract. We believe the revised manuscript more clearly represents our research. Thanks to the reviewer’s constructive comments.

Best regards,

Several critical concerns are listed as follows.

In the sentence line 157 authors said “antiviral ability of BCE was maintained regardless of the heat treatment”. However, in the figure 3, the line in yellow correspond to the heat-treated BCE and reach levels equivalents to the infected cells-not treated with BCE. Could the authors comments on it?

Same figure was inserted into the manuscript as figure 3 and 4. We feel deeply sorry for the confusion. An appropriate figure is inserted in figure 3 in revised manuscript. This figure shows that the heat treated BCE (yellow square line) and the autoclaved BCE (light blue triangle line) maintain a higher CI than the infected cell. Therefore, we could confirm the antiviral ability of BCE regardless of the heat treatment. Please see the Figure 3.

In the table 2 is not clear to which experimental set up correspond the “heat inactivated” treatment of the BCE. In any case, the combine treatment of the BCE, the trypsin and Proteinase K treatment also reduce the antiviral effect of the BCE. Could the authors comments on it?

Table 2 shows the effect of enzyme, not heat. Trypsin and Proteinase K are proteolytic enzymes. If the effect of BCE is due to a protein substance, it will disappear when Trypsin and Proteinase K are added. However, when the two enzymes were added, the effect of BCE did not disappear, indicating that the effect of BCE was due to the non-protein substance. In addition, the heat inactivated enzymes treatment without BCE seemed to be an unnecessary group in the above experiment, confirming that there was no anti-rotaviral effect. We removed the group from the table 2. We feel gratitude for your meaningful comments.

In the discussion the authors stated “found that a low molecular weight fraction of under 10 kDa was more effective than whole cell extract” nevertheless in the figure 4 there is not any significant analysis done. A commentary of the authors would be appreciated.

We modified the sentence to reflect your important comment. Please see the line #210-1.

The study missed the reproducibility of their experiments.

o Table 1: 8 wells were used as duplicated. It is missing what it is the difference in the wells. If they are only technical duplicates, why some wells get infected and some not? Equivalent comments for Figure 1a and figure 2: “n=3”.

The 8 well test is a repeat of the technical experiment. The constituents that make up the BCE are very diverse. Therefore, since the components of BCE could be treated not exactly equal concentration in wells, we set up a rather large number of 8 repetitions to see the overall trend. Figure 1A and Figure 2 was also performed with the 8 well test and measured with absorbance OD value of MTT assay. We revised the manuscript to reflect your comment and please see the line #146.

o Figure 1B, Figure 3 and figure 4 lack of any ecartype, is it meaning that they did only 1 well or test? How many times they repeated the experiment?

The experiments were performed with 1 test for observing the preliminary trend of anti-rotaviral activity in this short communication paper. Therefore, we modified the tone of the sentence in revised manuscript. Please see the line #210-2

- Regarding the percentage of inhibition of infections, represented with a Y-axis higher of 100% is not accurate.

The percentage of inhibition of infections was calculated with the MTT assay results, OD value. The inhibition of infection was expressed as the percent (%) of wells with uninfected cells among the total experimented wells. In some wells, infections due to viruses were inhibited, and when the OD value is higher than that of non-infected well cells, the inhibition of infection (%) can exceed 100%. This can also be caused by a slight seeding error. We additionally mentioned about the MTT assay in materials and methods section in revised manuscript. Please see the line #109-10.

- Figure 1: the authors do not comment the results regarding the high concentrations of BCE.

We have referred to the dose and incubation time dependency, including the highest concentration of 3 mg/ml in revised manuscript. Please see the line #138.

Specific comments:

- The colours of the graphics are not compatible with colour blind people. The authors should change the colours.

- Moreover, the authors should to include explanations about the sign used in the graphs for the different groups in the figure or in the legend figure, to avoid the colour-print of the paper. In the actual version, they only referred to the colours in the figure legend.

Thank you for the meaningful comments. In fact, we had not considered people who were colour blind first. However, in accordance with your advice, not only the colours but also the symbols are displayed in the legends. We hope that people with colour blindness could read the papers. Please see the line #163-6 and #177-81.

- In the lines 99 and 100 the units looks wrong: “50 L of virus solution”, is it correct?

We have corrected as you commented. Please see line #100-4

-In the “antiviral assay” concentrations should be clarified for the virus solution, and for the components of the final mix, rather than dilutions or volumes.

The concentration of the final mixture is indicated in the revised manuscript. Please see line 104.

- Figure 1 legend: “treatment” should be change by “exposure to the cells” or something equivalent.

We have corrected as you commented. Please see line #144-6

- Globally, the strategy to illustrate significant comparisons in figure 1A, figure 2 is not clear. Authors could improve it.

We have made new Figures to show dose and time dependency at the same time with statistical significance. Please see the Figure 1A, Figure 2, line #144-6 and #154-6.

- Line 146: “treated” should be change by “add” or something similar.

We have corrected as you commented. Please see line #149

- Line 171: type for “Figure”

We have corrected as you commented. Please see line #173.

- Line 203: “and allowed it to maintain a higher CI value than the negative control cells that had not been infected”

We have corrected as you commented. Please see line #205.

Round 2

Reviewer 2 Report

The comments and modifications done by the authors on the manuscript entitled: “ The anti-rotaviral activity of low molecular weight and non-proteinaceous substance from Bifidobacterium longum BORI cell extract” satisfy my concerns about their publication. The authors improved substantially the quality of the manuscript by implementing my comments.  

Author Response

We would like to express our appreciate for taking the time to share constructive feedbacks.